# MicroWorld: An Augmented-Reality Arabian App to Learn Atomic Space

**Mayda Alrige * , Hind Bitar , Waad Al-Suraihi, Kholoud Bawazeer and Ekram Al-Hazmi**

Faculty of Computing and Information Technology, King Abdulaziz University, Jeddah 21589, Saudi Arabia; hbitar@kau.edu.sa (H.B.); walsuraihi@stu.kau.edu.sa (W.A.-S.); kbawazeer0004@stu.kau.edu.sa (K.B.); eakhazmi0017@stu.kau.edu.sa (E.A.-H.)
* Correspondence: malraegi@kau.edu.sa

**Abstract:** The visualization of objects of an abstract nature has always been a challenge for chemistry learners. Thus, augmented reality (AR) and virtual reality (VR) have been heavily invested in as immersive learning methods for these concepts. This study targets the segment of the chemistry curriculum involving the chemical elements of the periodic table. For this purpose, we developed the AR educational tool called *MicroWorld*. This Arabic educational AR app was developed in unity with Vuforia SDK. Using *MicroWorld*, students can visualize chemical elements microstructures in 3D, see 3D models of the elements in their substantial forms, and combine two chemical elements to see how certain chemical compounds can be formed. In this work, *MicroWorld*'s usability was evaluated by junior high school students and chemistry teachers using the Arabic system usability scale (A-SUS). The A-SUS average score was 71.5 for junior high school students, while the scale for teachers reached 76. This research aims to design, develop, and evaluate the AR app, *MicroWorld*. This app was built and evaluated through the lens of the design science research paradigm.

**Keywords:** augmented reality; immersive learning; periodic table; chemistry





## 1. Introduction

Junior high school students still consider chemistry one of the most difficult courses to grasp and learn its abstract concepts. To fully comprehend these concepts, they must integrate information from the macroscopic, microscopic, and symbolic domains of the discipline. Abstract concepts, such as atoms or special representation by the element, are most forgettable to junior high school students. Lessons that require students to imagine chemical elements and compounds across micro- or macroworlds, can be extremely challenging [1].

Many emerging technologies have been integrated into the learning process to foster certain educational concepts. Technologies delving into different kinds of augmented, visual, and mixed realities (AR/VR/MR) are referred to as immersive methods of learning. AR/VR/MR are interactive instruments of immersive learning [2].

Previous studies have shown that AR can stimulate high levels of engagement [2]. Furthermore, this technology has been found to be useful when the studied phenomena cannot be simulated in reality, such as phenomena related to the solar system [3,4]. One systematic review found that the use of AR in Science, Technology, Engineering, and Mathematics (STEM) education can foster students' conceptual understanding [5].

With the global pandemic, AR tools are now much more feasible and helpful in educational environments for the smooth integration of immersive learning methods, online education, and hybrid education. In this research paper, we developed and evaluated an AR Arabic educational tool called *MicroWorld*. This tool aims to foster a better understanding of the chemical elements in the periodic table in Arabic-speaking junior high school students. The *MicroWorld* app offers the following features: (1) Arabic interfaces; (2) images of each chemical element in its substantial nature; (3) three-dimensional models

of the uses of the chemical elements in real life; and (4) models of chemical element atoms to better help students imagine the spatial representations of the atoms and memorize atomic numbers. *MicroWorld* also uses the smartphone camera to detect a marker from the current view. In addition, if the app detects two elements that compose a chemical compound, it shows a three-dimensional (3D) model of that compound. *MicroWorld* was evaluated on the basis of usability by junior high school students and chemistry teachers using the Arabic system usability scale (A-SUS). The average A-SUS score was 75 for junior high school students, while the average score for teachers came to 76. These scores are considered high, and indicate that *MicroWorld* is technically usable and can help chemistry students build their understanding of chemical elements and compounds, and imagine atomic decomposition in space [6]. To ensure that this app meets its expected results, unit testing was conducted; all the scenarios successfully passed the testing. We argue that the *MicroWorld* app is a promising tool that can facilitate the learning process for both learners and teachers.

## 2. Literature Review

The use of computers and communication technologies has been rapidly increasing, especially with the increased use of smartphone apps and AR technology targeting teachers and students in the learning environment [7]. Previous studies suggest that AR has the capability to attract student's attention and help them learn within a fun environment, especially in the case of remote learning. To determine if AR/VR technologies could be beneficial to the technological transformation of remote learning, a systematic umbrella review of 68 articles [8] identified 30 chemical elements that contained either AR or VR interventions in remote learning in higher education that impacted either academic performance or engagement. Among these research papers, 24 showed measured impact on performance and six described measured impacts on engagement.

The use of AR in the education sector has been increasing rapidly because the technology allows users to insert virtual objects into a real-world view using a device camera and screen. This ability has proven useful for courses that teach atomic and molecule structures or other concepts of an abstract nature. According to Cai and his colleagues, AR is most applicable in the following two cases: (1) when the phenomenon cannot be simulated in reality, such as the solar system [3]; (2) when real experiments have conspicuous shortcomings, such as the convex imaging experiment [4], as it is dangerous to keep a lighted candle in a classroom. Many attempts have been made to explore the potential of augmented reality in STEM educational topics, such as geometry [9] and computer science [10].

For chemistry teaching, it has been proven that academic performance can be improved by incorporating an AR micro-structure learning tool [1]. AR has been incorporated to foster different concepts; for example, ARLab shows students 3D models of lab flasks with important descriptions of each flask and its use. Aimed at high school students, *ARLab* helps students learn chemistry glassware, using realistic 3D models of both volumetric and graduated glassware. *ARLab*'s main goal is to teach students important concepts, such as correct measurement and glassware applications in a chemical reactions lab [11].

In addition, Maier and Klinker developed the *Augmented Chemical Reaction* app that shows the 3D spatial structure of molecules as well as the dynamics of the atoms in and between molecules. However, users do not interact with the commonly used 3D user interface via mice or keyboards to move virtual objects. Rather, the developers designed a direct manipulation user interface using an augmented reality technique. This new method enables users to better understand the spatial structures of the shown geometries. Maier and Klinker's AR tool was targeted at students from elementary and high school [12]. Another study presented an app for students that showed the molecular structures of the different chemical elements [13]. Also, an AR system was introduced for teaching inorganic chemistry at the university-level [13]. This proposed system used inexpensive cameras and open-source software to set up a collaborative environment that supported several groups of students interacting with material and compound structures [1].

To the best of our knowledge, there is limited implementation and usage of AR technology to assist Arabic-speaking students and teachers during chemistry lessons. Most of the developed AR tools were designed to support students in other courses, such as Arabic courses [14], courses targeting impaired/disabled students [15], interactive writing courses [16], and early childhood development courses [17]. The only AR app we could find targeting Arabic speakers was developed and tested by Ewais and Troyer [18], who designed and developed an AR application to assist female elementary school students in Palestine in learning the reactions among atoms and molecules. The testing focused on the students' attitude changes after using the application. The analysis showed that female students had a positive attitude regarding the usage of AR applications during the learning process.

Our research targets the teaching of the "Chemical Elements and Compounds in the Periodic Table" segment of the chemistry curriculum, which begins in seventh grade and continues to a detailed level in ninth grade. All descriptions and explanations of the periodic table are in Arabic.

### 3. Materials and Methods

We followed the design science research (DSR) framework to build and evaluate the *MicroWorld* app. DSR is better suited where the goal is to solve practical problems by designing and building artifacts in an iterative process. This process can be better viewed as the embodiment of three related activity cycles: relevance, rigor, and design. The relevance cycle initiates the DSR project by addressing the contextual environment of the research project. In this cycle, the authors consulted the apps available in both Google Play and the Apple store that incorporate augmented reality to assist in chemistry learning. The rigor cycle defines the knowledge base of the scientific foundations in the form of theoretical models/theories, experience, and expertise. It aims to ground the DSR artifact to the knowledge base and define its novelty. During this cycle, we supply the periodic table materials, including the basic chemical elements and their information, such as atomic number and molecular space. We also conducted a literature review for all AR assistive tools that have been developed and evaluated to assist in chemistry teaching.

The design cycle iterates between the essential activities of building and evaluating. This cycle is thoroughly explained in the following two subsections. The three main cycles of DSR are illustrated in Figure 1.

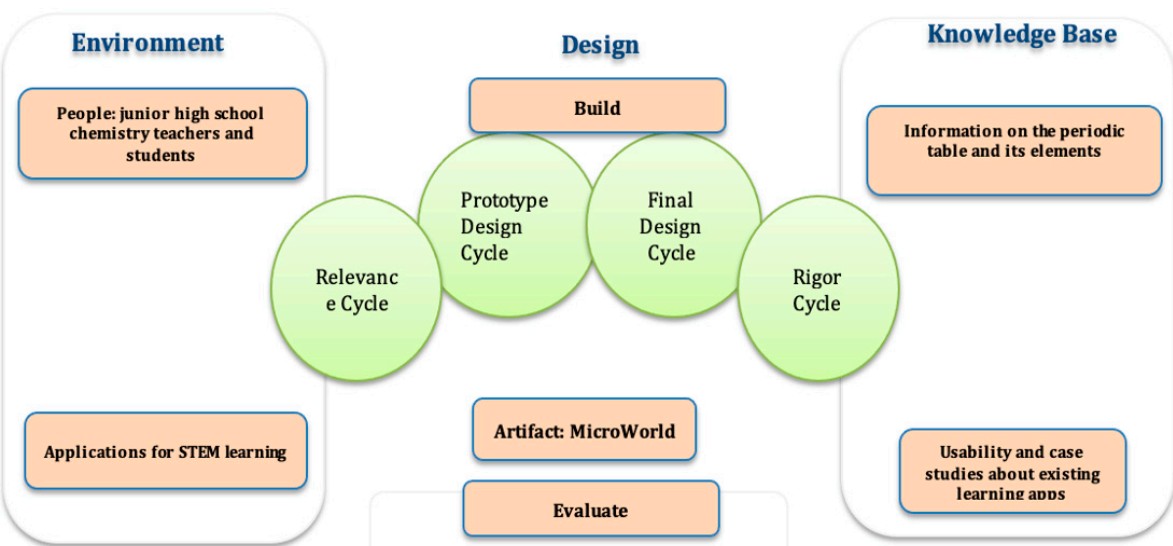

**Figure 1.** DSR cycle to build and evaluate the *MicroWorld* app.

### 3.1. The Artifact MicroWorld App: Building Activity

The *MicroWorld* app was built using the Unity engine and Android Studio. First, we created a database on the Vuforia platform. Vuforia is an augmented reality software development kit (SDK) for mobile devices that enables the creation of augmented reality applications. It uses computer vision technology to recognize and track planar images and 3D objects in real time. This image registration capability enables developers to position and orient virtual objects, such as 3D models and other media, in relation to real-world objects when they are viewed through the camera of a mobile device. The virtual object then tracks the position and orientation of the image in real time so that the viewer's perspective on the object corresponds with the perspective on the target [19].

We utilized the Vuforia platform to upload the targets as images of the symbols of 15 chemical elements taken from the periodic table. We included these 15 basic chemical elements and five chemical compounds in our database. We then downloaded them as Unity packages to be accessible by the mobile application, as shown in Figure 2.

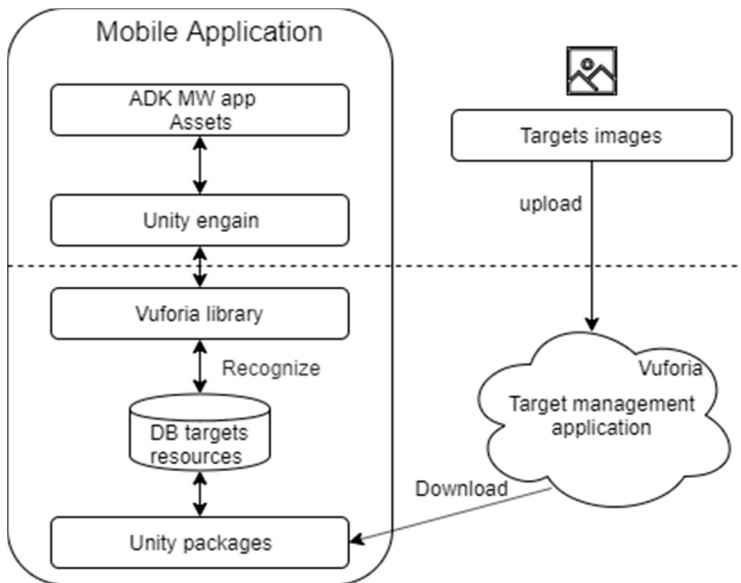

**Figure 2.** The architecture of the *MicroWorld* app.

The Arabic interface of the chemical elements contains four buttonsË (1) showing the image of the raw chemical element, (2) showing a three-dimensional model of the element's uses, (3) displaying element information, and (4) showing the atoms of this element. The workflows of *MicroWorld* is shown in Figure 3, with the final interfaces of the four main features for each chemical element shown in Figure 4.

### 3.2. The Artifact MicroWorld App: Evaluation Activity

The evaluation in design science research can be done from two perspectives: a socio-technical perspective and/or a technical perspective. In this research, we evaluated the utility of *MicroWorld* app from both perspectives. Two iterations were done from a socio-technical perspective. The first iteration was meant to evaluate the usability and learnability of the app's prototype among chemistry teachers and junior high school students in the city of Jeddah in Saudi Arabia. We use convenience sampling to distribute the usability survey along with a link to the screen recording of the *MicroWorld* intervention.

The survey link along with the prototype was sent via two social-media channels: WhatsApp and Facebook. The link was sent on the 13 April 2020. Having the students experiment with *MicroWorld* in the classroom was not an option since all schools were giving their classes online through a distance education platform at the time. The interfaces of the prototype are shown in Table 1.

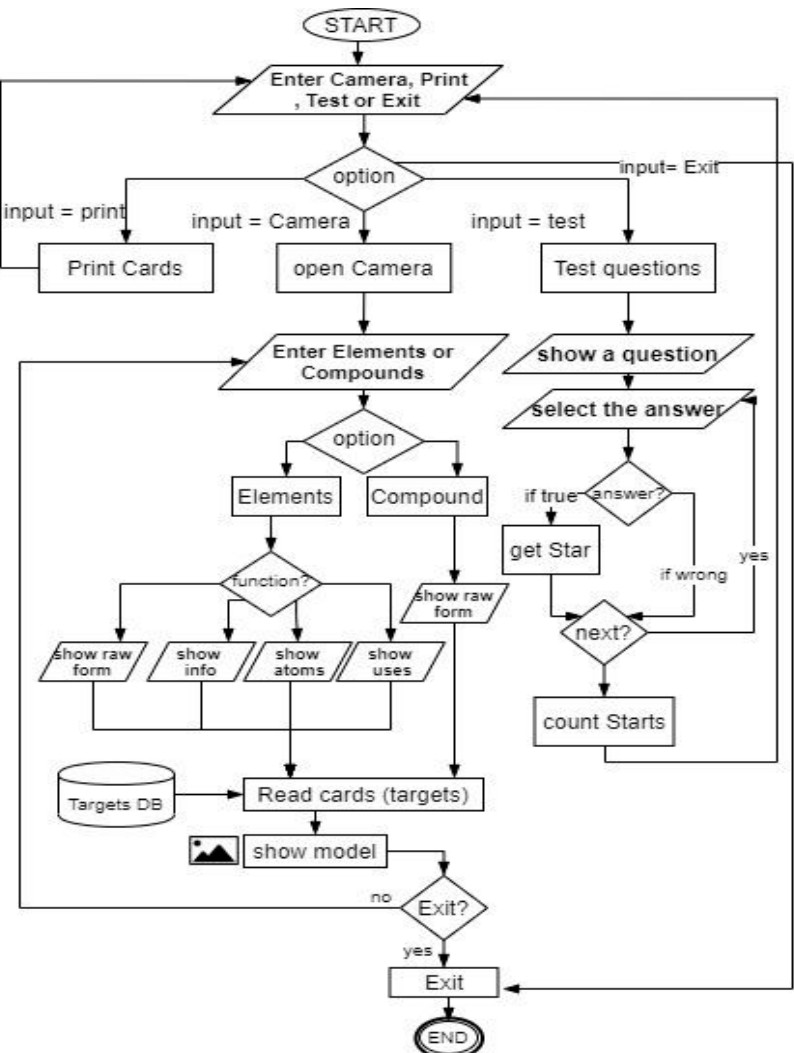

**Figure 3.** Flowchart of the *MicroWorld* app.

To evaluate the usability of *MicroWorld* in the first iteration, we adopted the Arabic-System Usability Scale (A-SUS) by AlGhannam and his colleagues [20]. This is an Arabic translation of the widely used System Usability Scale (SUS), a 10-question questionnaire used to assess the usability of various systems and applications. SUS is known for its wide acceptance, robustness, and reliability, even with a small number of participants. In the Arabic version, five professional translators translated SUS from English to Arabic. Then, the translations were presented to an Arabic linguistic professor for further examination and assessment until an agreement was reached. To assure accurate translation, the Arabic questions were back translated to English, and the result was contrasted with the original English questions. A-SUS was evaluated empirically by students in the Department of Communication Disorders Sciences in an Arabic university and was seen to have an acceptable result in measuring the usability of mobile applications. The A-SUS was used by [21] to evaluate the usability of an augmented reality app for teaching chemistry in a secondary school. The experiment was conducted by posting a link to the prototype of *MicroWorld* on two social media channels. Then, the participants were asked to fill out the A-SUS questionnaire to consolidate and conclude the prototype's evaluation.

The second iteration of the evaluation was to improve the prototype based on the feedback we obtained from the first iteration. After developing the front-end and back-end of *MicroWorld*, we evaluated the app from a technical perspective by conducting unit-testing, technical testing that focuses on the smallest units in the system [21]. For our unit

testing, we tested the technical performance of every single and smallest function in the *MicroWorld* app to ensure that it met the expected requirements.

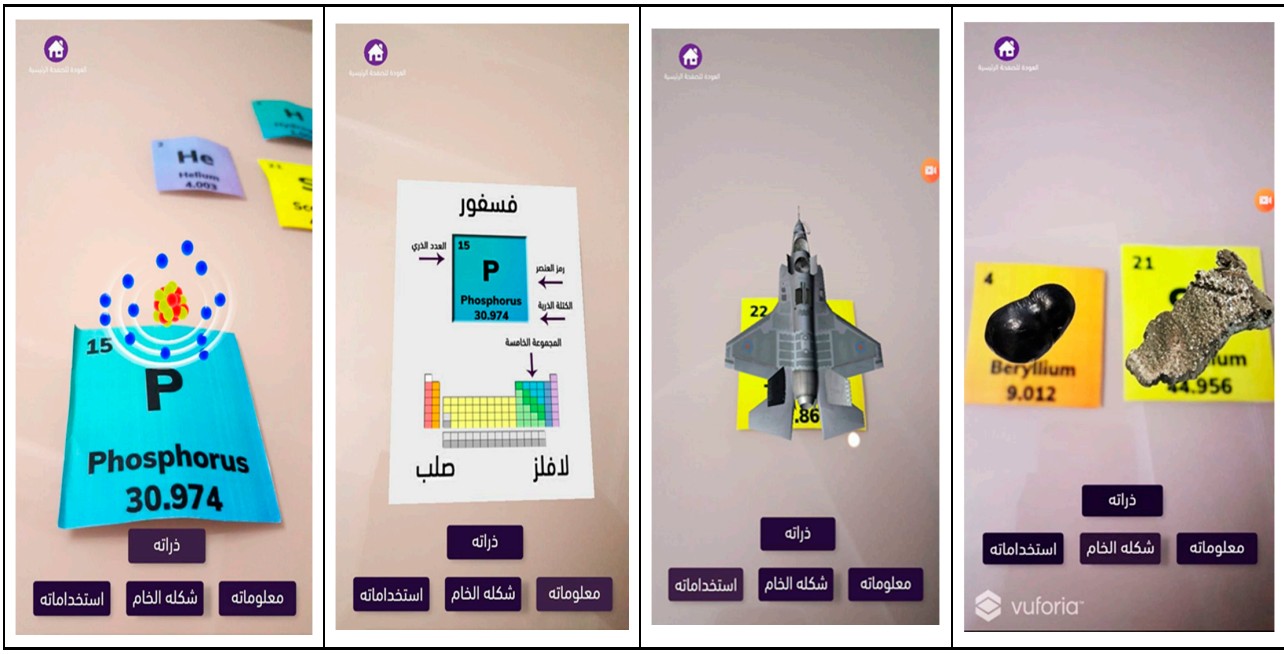

**Figure 4.** Phosphorus atoms in 3D, general information, raw material in 3D, and its uses in 3D.

**Table 1.** The interfaces of the *MicroWorld* initial prototype.

| Interfaces | Descriptions |
|---|---|
| 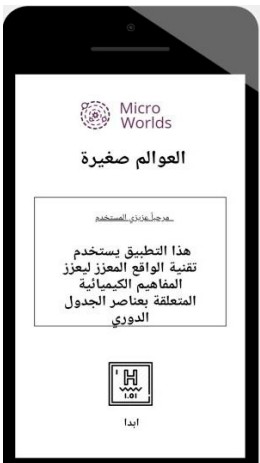 | The home page has simple information about the *MicroWorld* application and one button to turn on the camera. |
| 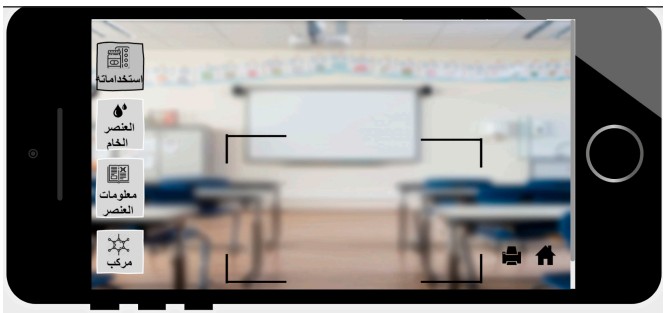 | The mobile camera will open and the user will direct the camera to the chemical element card so that the app detects the target (card). |

**Table 1.** *Cont.*

| Interfaces | Descriptions |
| --- | --- |
| 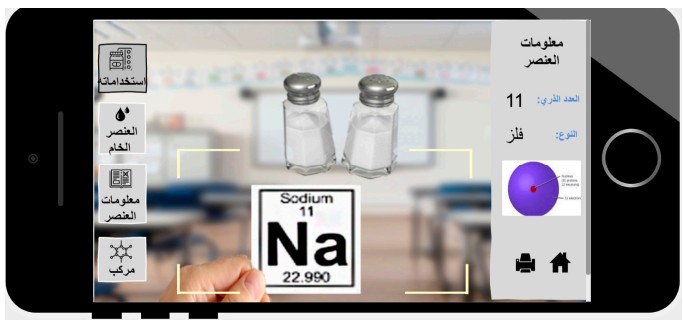 | After the app detects the target, the user can choose any function. When the user clicks on the chemical element uses button, the app will display a 3D model of these uses. |
| 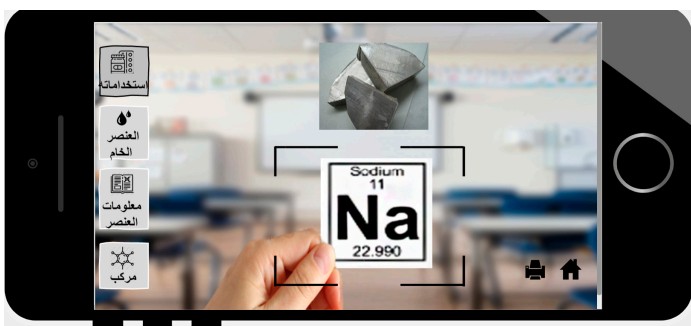 | When the user clicks on the button for the raw form of the chemical element, the app will display an image of this chemical element in its natural (raw) form. |
| 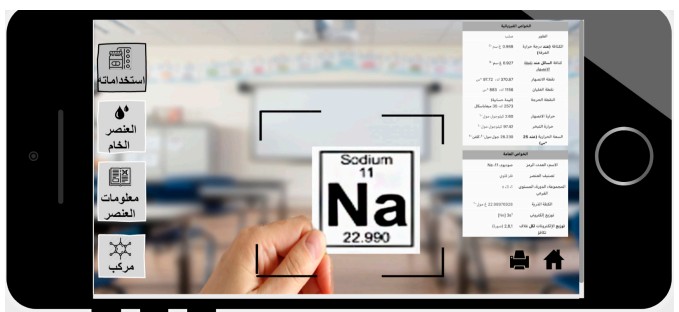 | When the user clicks on the button for chemical element information, the app will display its information. |
| 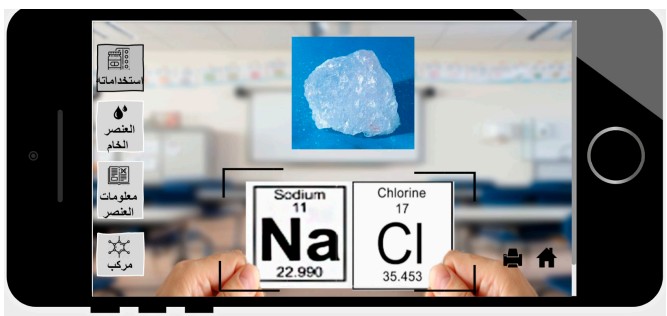 | When the user clicks on the chemical compound in real life button and the application detects any two targets (chemical elements cards saved as targets), the app displays a 3D model of the compound. |

## 4. Results

In the first iteration, about 40 participants completed the survey to evaluate the usability of *MicroWorld*. The survey consisted of three sections: The first section contained demographic questions, the second contained the ten A-SUS questions, and the third section contained feedback questions. The system's usability score was calculated by two groups of users: chemistry teachers and junior high school students. The A-SUS score for students came to 75 (see Table A1), while the A-SUS score for the teachers came to 76 (see Table A2). This score is above the average score of 68. This SUS score for any software indicates that it is technically usable and has no learnability issues [22]. For *MicroWorld*,

the A-SUS score for both teachers and students are above the average. This indicated that the prototype of *MicroWorld* is technically usable and learnable.

In addition, the prototype evaluation highlighted some of the issues that we can address before finalizing the back-end of *MicroWorld*. Almost all the improvement points obtained during this iteration in the socio-technical perspective were addressed. For example, one of the improvements points we obtained through the feedback questions was to add quizzes at the end of each session. These quizzes help the students to measure their understanding of the chemical elements and compounds. In another world, how much *MicroWorld* would have helped students foster their understanding and reinforce the knowledge gained while experimenting with the app.

The second iteration of the evaluation is meant to measure *MicroWorld* from a technical perspective. Thus, unit testing was conducted to assure that the actual results met the expected results [21]. This iteration of evaluation is performed to assure that *MicroWorld* has no performance issues. Tables A1– A4 outline unit testing results for *MicroWorld* app's features. All results of all unit tests were positive.

## 5. Discussion

*MicroWorld* is an AR learning tool developed to facilitate and foster the learning process of the chemical element's atomic structures and substantial nature through the use of 3D models. The usability of the Android-based app was tested among junior high school students and chemistry teachers. Initial findings of the pilot study suggest that *MicroWorld* is technically usable and learnable for seventh grade students. The utility and the novelty of *MicroWorld* app are essential to highlight. We argue that applying *MicroWorld* would indeed foster the learning process of chemical concepts for Arabian-speaking junior high school students with a significant reduced amount of time and effort during learning process. This app would help solve one of the difficulties that seventh grade learners face during the school year, which is learning about chemical element's atomic structures and substantial nature. The *MicroWorld* app would empower students during their learning journey by aiding them to fully understand the periodic table in chemistry courses using AR technique.

Incorporating AR tools into education has been widely studied and proven useful for STEM learning. Cutting edge technologies such as AR, VR, and MR, strongly support immersive methods of learning [2], especially during the COVID-19 pandemic. AR, in particular, has shown to improve the learning of individual students, motivate them, and assist in organized teamwork and group cooperation [1,2]. This particular time paves the way for many innovative technologies to come along and foster the educational process more than ever before.

Since the pandemic started at the beginning of 2020, many countries have opted for distance learning for K–12 education and colleges, with some variations. In Saudi Arabia, the Ministry of Education (MOE) utilized the Madrasati platform as a new gateway for distance learning during the pandemic. Since most of the educational systems worldwide use their own distance learning platforms, incorporating such cutting-edge technologies as *MicroWorld* is likely to be supported. AR tools such as *MicroWorld* can be effectively integrated to enhance the online educational experience and foster educational concepts.

Reluctance to use and learn to use these technologies will not be as much of an issue as it was before the pandemic [23]. The feedback we obtained had indicated that some teachers might be reluctant to use such apps because they did not have the time to learn and adopt technologies and that they would rather use the traditional way of teaching. We argue that these mindsets may have changed during the global pandemic. Many teachers are becoming more willing to get trained and adopt cutting-edge technologies to improve the educational system and assist students in gaining the assigned knowledge.

## 6. Conclusions, Limitations, and Future Work

The *MicroWorld* app is a useful assisting tool to help teachers in transferring new required knowledge to their students, and aids students in fully understanding the periodic table in chemistry courses targeting Arabian students and teachers. The first and the second iteration in the evaluation phase show that this app is a promising solution for Arabian chemistry students and teachers. The A-SUS score for the app indicates that it is technically usable and has no learnability issues. Thus, there are not any technical issues with the app interfaces. The app is consistent, easy to use without the need for any technical assistant around. Students and teachers have indicated that they would learn how to use it very quickly without any specific technical background. Also, the unit testing results indicate that there are not any technical and performance issues of the IT artifact, *MicroWorld* app. This app would also facilitate the educational process, especially when online learning is required due to the COVID-19 pandemic. The use of AR technology in developing the app would assist the students to learn and visualize chemical elements molecule space in a fun, enjoyable way.

The main limitations of this research are in its evaluation phase. Due to the current conditions of the COVID-19 pandemic, when we evaluated the final app, we could only conduct unit testing; more evaluation must be performed to ensure the app's utility and efficiency. In the future, another evaluation stage from a socio-technical perspective will be added.

To evaluate the efficiency and efficacy of *MicroWorld*, we plan to conduct an experimental study in a class setting in the near future once all schools reopen for students. We will conduct a quasi-experimental study targeting seventh grade students in a real environment. For this testing, the students will be sorted into two groups: the intervention (*MicroWorld*) group and the control group. The control group will keep using the standard method of learning, while the intervention group will use the *MicroWorld* app on top of that. To compare the groups and measure the degree of change occurring as a result of treatment, the two groups will receive the same test after the same period of time. Then, we will gather their results to compare the performances of the two groups [24]. For the test, the two groups will be given a set of questions from the California Chemistry Standards Tests [25] to measure the effect of the app's use on the students' academic performances regarding the chemistry concepts being taught. We are also planning to extend the features of *MicroWorld* as follows: (1) Add a chemical lab using VR technology, and (2) seek to cooperate with the Ministry of Education to get permission to apply *MicroWorld* in all schools, in cooperation with the national educational portals.

**Author Contributions:** Conceptualization, K.B. and W.A.-S.; methodology, M.A.; validation, M.A. and K.B.; resources, H.B.; software, E.A.-H.; writing—original draft preparation, M.A.; All authors have read and agreed to the published version of the manuscript.

**Funding:** This research received no external funding.

**Institutional Review Board Statement:** Ethical review and approval were waived for this study, due to pandemic. The study was not conducted in classroom as an experimental study. Rather, it was conducted as a cross-sectional study in a pilot test.

**Informed Consent Statement:** Not applicable.

**Data Availability Statement:** The data is attached with manuscript.

**Conflicts of Interest:** The authors declare no conflict of interest.

# Appendix A

**Table A1.** Unit testing cases of showing the chemical element in substantial nature.

| Unit Test ID | Description | Input | Expected Result | Actual Result | Pass /Fail | Remark |
|---|---|---|---|---|---|---|
| 1 | The user directs the camera to an electronic image of a known card (not printed). MicroWorld should display an image of its element in its raw form. | The user directs the camera to an electronic image of a known card (not printed). MicroWorld should display an image of its element in its raw form. | Image of raw carbon is displayed | Image of raw carbon is displayed | Pass | Positive 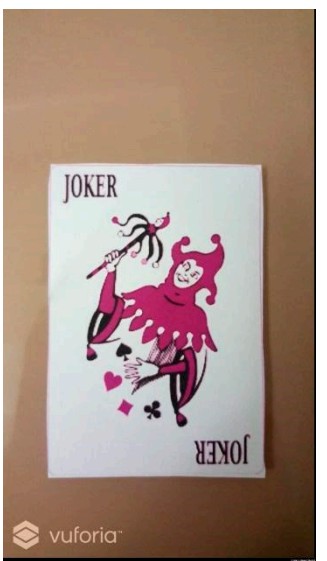 |
| 2 | User directs the camera to something unknown, or an unknown and different card. MicroWorld should not display anything. | Unknown card or target | Nothing is displayed | Nothing is displayed | Pass | Positive |

**Table A2.** Unit testing cases of showing a 3D model of one use of the chemical element.

| Unit Test ID | Description | Input | Expected Result | Actual Result | Pass /Fail | Remark |
|---|---|---|---|---|---|---|
| 3 | The user directs the camera to a known card. *MicroWorld* should display a 3D model of one of the element's uses. | Scandium element target | Displays a 3D model of a bicycle, one of the uses of scandium | Displays a 3D model of a bicycle, one of the uses of Scandium | Pass | Positive  |

**Table A3.** Unit testing case of unknown card.

| Unit Test ID | Description | Input | Expected Result | Actual Result | Pass/ Fail | Remark |
|---|---|---|---|---|---|---|
| 4 | User directs the camera to something unknown or unknown and different card. *MicroWorld* should not display anything. | Unknown card or target | Nothing is displayed | Nothing is displayed | Pass | Positive  |

**Table A3.** *Cont.*

| Unit Test ID | Description | Input | Expected Result | Actual Result | Pass/ Fail | Remark |
|---|---|---|---|---|---|---|
| 5 | The user directs the camera to a known card. *MicroWorld* should display a model of the element's atomic structure. | Hydrogen element target | Model of a hydrogen atom is displayed | Model of a hydrogen atom is displayed. | Pass | Positive  |
| 6 | The user directs the camera to a known card. *MicroWorld* should display a model of the element's atoms. | Helium element target | Model of a helium atom is displayed | Model of a helium atom is displayed | Pass | Positive  |

**Table A4.** Unit testing cases of showing a 3D model of one use of the chemical compound when directing the camera toward its two main chemical elements.

| Unit Test ID | Description | Input | Expected Result | Actual Result | Pass /Fail | Remark |
|---|---|---|---|---|---|---|
| 7 | The user directs the camera to two cards (targets). *MicroWorld* should display a model of the result of combining their chemical elements. | Oxygen and hydrogen elements (targets) | A model of water, the result of combining these chemical elements, is displayed. | A model of water, the result of combining these chemical elements, is displayed. | Pass | Positive 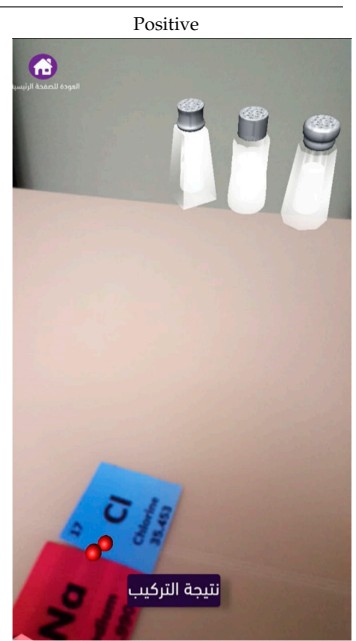 |
| 8 | The user directs the camera to another two cards (targets). *MicroWorld* should display a model of the result of combining their chemical elements. | Sodium and chlorine elements (targets) | A model of salt, the result of composing these chemical elements, is displayed. | A model of salt, the result of composing these chemical elements, is displayed. | Pass | Positive |

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
