# Peer review of "MicroWorld: An Augmented-Reality Arabian App to Learn Atomic Space"

_technologies, doi:10.3390/technologies9030053_

Round 1
Reviewer 1 Report
The article is very interesting. The use of augmented reality applications in education is a very topical issue. AR MicroWord is similar to AR Elements 4D. The design of the research is not complete, it would be appropriate to supplement the data on when (date) the research was carried out. It would be appropriate to describe in more detail the implementation and evaluation of the research. After completing this information, I recommend publishing the article.
Author Response
Please see the attachment.
Point 1: The article is very interesting. The use of augmented reality applications in education is a very topical issue. AR MicroWord is similar to AR Elements 4D. The design of the research is not complete, it would be appropriate to supplement the data on when (date) the research was carried out.
Response 1: Thanks for the comment. The date of first iteration of the evaluation has been added to the manuscript page 7 under the Evaluation Activity section. All the changes are tracked). We have sent the survey of A-SUS on April 13th, 2020.
Point 2: It would be appropriate to describe in more detail the implementation and evaluation of the research. After completing this information, I recommend publishing the article.
Response 2: Thanks for the comment. More details have been added to the evaluation activity in both first and second iteration of evaluation. These details include:
- Datils of the purpose of the two iterations
- Details on how to interprets SUS results and what does it mean to have a SUS score equal to 75 (by students) and 76 (by teachers).

Reviewer 2 Report
The article that is presented is of interest in the educational area. The project has been worked with scientific rigor, but in my opinion, it is not shown that the MicroWorld application is a promising tool that can facilitate the learning process for both students and teachers.
The results section is to present data, as well as to comment on and interpret them, this section does not contain this information so a synthesis and organization effort must be made to present them and give a proven response to the proposed evaluation objectives.
It is necessary to include in the results section data that refer above all to the evaluation from the sociotechnical perspective, specifically to the learning capacity of the application prototype according to the students and teachers.
The conclusions can be improved in the light of the results that are included by going deeper into the details of the evaluations carried out.
Author Response
Please see the attachment.
Point 1: The results section is to present data, as well as to comment on and interpret them, this section does not contain this information so a synthesis and organization effort must be made to present them and give a proven response to the proposed evaluation objectives.
It is necessary to include in the results section data that refer above all to the evaluation from the sociotechnical perspective, specifically to the learning capacity of the application prototype according to the students and teachers.
Response 1: Dear reviewer, this is a very valuable comment, we appreciate it. The results were edited and synthesized in light of the two main iterations. The effect of using the app on academic performance among many other learning outcomes will be investigated in future work once schools reopen in Saudi Arabia. The initial plan was to conduct two-group quasi experimental study and measure the academic performance of students with and without using Microworld. The detail of this future direction is moved to the discussion part. As for this research project, because of the pandemic, we were able to measure only the usability of the prototype through online survey and improve the whole app based on users’ feedback.
Point 2: The conclusions can be improved in the light of the results that are included by going deeper into the details of the evaluations carried out.
Response 2: Dear reviewer, this is also a very valuable comment, we appreciate it. We addressed your comment as shown in the Conclusions, Limitations and future work section. We add more details regarding the performed evaluation, as well as the future evaluation plan.
